# The Enhancement of CO_2_ and CH_4_ Capture on Activated Carbon with Different Degrees of Burn-Off and Surface Chemistry

**DOI:** 10.3390/molecules28145433

**Published:** 2023-07-15

**Authors:** Supawan Inthawong, Atichat Wongkoblap, Worapot Intomya, Chaiyot Tangsathitkulchai

**Affiliations:** School of Chemical Engineering, Institute of Engineering, Suranaree University of Technology, Nakhon Ratchasima 30000, Thailand; supawan.int@gmail.com (S.I.); worapojinthomya@gmail.com (W.I.); chaiyot@sut.ac.th (C.T.)

**Keywords:** activated carbon, adsorption, defective surface, GCMC, perfect surface

## Abstract

Activated carbon derived from longan seeds in our laboratory and commercial activated carbon are used to investigate the adsorption of methane (CH_4_) and carbon dioxide (CO_2_). The adsorption capacity for activated carbon from longan seeds is greater than commercial activated carbon due to the greater BET area and micropore volume. Increasing the degree of burn-off can enhance the adsorption of CO_2_ at 273 K from 4 mmol/g to 4.2 and 4.8 mmol/g at 1000 mbar without burn-off, to 19 and 26% with burn-off, respectively. This is because an increase in the degree of burn-off increases the surface chemistry or concentration of functional groups. In the investigation of the effect of the hydroxyl group on the adsorption of CO_2_ and CH_4_ at 273 K, it is found that the maximum adsorption capacity of CO_2_ at 5000 mbar is about 6.4 and 8 mmol/g for cases without and with hydroxyl groups contained on the carbon surfaces. The opposite behavior can be observed in the case of methane, this is due to the stronger electrostatic interaction between the hydroxyl group and carbon dioxide. The simulation results obtained from a Monte Carlo simulation method can be used to support the mechanism in this investigation. Iron oxide is added on carbon surfaces with different concentrations to reveal the effects of ferric compounds on the adsorption of CO_2_. Iron at a concentration of about 1% on the surface can improve the adsorption capacity. However, excessive amounts of iron led to a limited adsorption capacity. The simulation result shows similar findings to the experimental data. The findings of this study will contribute to the progress of gas separation technologies, paving the way for long-term solutions to climate change and greenhouse gas emissions.

## 1. Introduction

The gas adsorption process is extensively utilized in a variety of gas separation [1,2] and purification applications [3], as well as methane storage [4] and environmental protection. Global warming caused by greenhouse gases is one of the greatest scientific interests. Some of these gases are sulfur hexafluoride [5], carbon dioxide, methane, and nitrous oxide [6,7]. The rising levels of carbon dioxide and methane in the Earth’s atmosphere have become of the utmost importance due to their substantial contributions to climate change. Developing effective strategies for the separation and purification of CO_2_ and CH_4_ is crucial for mitigating greenhouse gas emissions and addressing global warming.

A novel technology for minimizing carbon dioxide emissions at their source is called enhanced coal bed methane recovery, in which an adsorption process is applied to recover methane in the underground coal bed using carbon dioxide injection. This occurs because carbon dioxide is preferentially adsorbed onto coal over methane [7,8], therefore, the coal will capture carbon dioxide and release methane as a useful natural gas. This process allows a reduction in carbon dioxide and also makes carbon dioxide storage economically feasible [7]. In recent years, methane adsorption in porous solids has been investigated for the purpose of methane storage and transportation via adsorbed natural gas (ANG) instead of the original compressed natural gas (CNG) [4,9]. Because methane is the major component of natural gas and it has low critical temperature, it cannot be liquefied by an operation at room temperature [10]. The original CNG requires an expensive multi-stage compression up to high pressure at 25 MPa that carries the risk of explosion; the ANG method operates at 3–4 MPa for the same capacity of methane storage [9,10].

Activated carbon, known for its exceptional adsorption properties because of its huge micropore and mesopore volumes as well as its high surface area, is commonly utilized in the adsorption process. It has emerged as a promising candidate for CO_2_ and CH_4_ capture and removal. However, the performance of activated carbon in gas capture processes can be substantially impacted by its surface heterogeneity, which results from differences in surface chemistry and morphology.

In recent years, extensive research has been conducted on the effects of surface modifications on the CO_2_ and CH_4_ adsorption behavior of activated carbon. The modifications are intended to customize the surface properties of activated carbon in order to optimize its performance for specific gas separation applications. By incorporating functional groups such as amines [11,12] and hydroxyls [13], the surface chemistry of AC can be modified to improve its interaction with CO_2_ and CH_4_. Incorporating metal impregnation [14] and iron oxide impregnation [15,16] onto the AC surface can provide active sites for adsorption, thereby enhancing the separation efficiency.

Previous studies have explored the impact of activated carbon surface modification on the adsorption behavior of carbon dioxide and methane with experimentation as mentioned. This paper investigates the effects of surface heterogeneity on CO_2_ and CH_4_ capture on activated carbon through a combination of experimental studies and simulation techniques. The integration of experimental and computational methodologies provides a potent way to investigate the impact of surface heterogeneity on CO_2_ and CH_4_ capture on activated carbon. The activated carbon samples were prepared with controlled surface heterogeneity, including defect surfaces, functional groups, and metal loadings. Simultaneously, the adsorption behavior of CO_2_ and CH_4_ on the heterogeneous activated carbon surfaces was simulated using a grand canonical Monte Carlo (GCMC) method. The GCMC simulations allowed for the investigation of the adsorption isotherms and diffusion dynamics of the gases within the activated carbon. This study addresses the limitations of infinite pore models commonly used to simulate porous carbon, such as infinite slit pores [17,18] or infinite cylindrical pores [19]. However, the infinite pore model is far too perfect to accurately represent the adsorption behavior of activated carbon, which has a finite length and contains functional groups or defects on the basal graphene layers [20,21]. Instead, a more realistic approach is taken by modeling methane and carbon dioxide adsorption using a real carbon pore with a finite length and graphene layers arranged in a hexagonal configuration [22]. The model includes defects, functional groups, and iron loadings. GCMC simulation is employed to determine the adsorption isotherms of carbon dioxide and methane at different temperatures and pore widths. The simulations are performed under varying conditions, such as different temperatures and different surfaces of the activated carbon, to investigate the actual capture scenarios.

This study aims to unravel the complex relationship between surface heterogeneity and the capture of CO_2_ and CH_4_ on activated carbon. The results of this study have the potential to influence the development of activated carbon materials. Furthermore, the findings of this study may contribute to the progress of additional adsorption-based gas separation technologies, paving the way for long-term solutions to climate change and greenhouse gas emissions.

## 2. Results and Discussion

We will start our discussion by presenting the experimental data, and then present the simulation results for carbon dioxide adsorption in activated carbon with a defective surface and perfect surface that were obtained from the GCMC results. We will compare the effects of pore width and temperature on the adsorption isotherms between the defective and perfect surfaces. Then, the adsorption of methane on the perfect and defective surfaces at 273 and 300 K will be discussed. Finally, the effects of the functional groups on the adsorption isotherms of methane and carbon dioxide will be presented.

### 2.1. Adsorption of Carbon Dioxide and Methane on Activated Carbon

#### 2.1.1. Adsorption Isotherms of CO_2_ and CH_4_ on Activated Carbon

The experimental adsorption isotherms of CO_2_ and CH_4_ on LACO at 273 and 300 K are shown in Figure 1. It appears that the activated carbon samples are better at adsorbing CO_2_ than CH_4_ since they adsorb more CO_2_ than CH_4_ under the same pressure and temperature conditions. This could be because the molecules of the two adsorbates are different sizes. With a kinetic diameter of 0.33 nm, CO_2_ molecules can easily pass through the majority of the samples’ pores. Contrarily, CH_4_ molecules have a kinetic diameter of 0.38 nm [23], making it more difficult for them to pass through the pores [24]. Obviously, the adsorption process is dependent not only on equilibrium, but also on the adsorbate’s accessibility to the porous structures of the material.

Figure 2 depicts the impact of the adsorption temperature for CO_2_ adsorption in (a) and CH_4_ adsorption in (b). It is clear that for the equilibrium adsorption isotherms of LACO, the amount adsorbed increases with decreasing temperature for both CO_2_ and CH_4_. In general, adsorption processes can be broken down into two categories: chemical adsorption and physical adsorption. The classification of an adsorption process is determined by the force that is exerted between the adsorbate and the adsorbent [25,26]. Physical forces are produced between the molecules of the adsorbent and the adsorbate in this work. This physical adsorption procedure can be regarded as exothermic as a result.

#### 2.1.2. Adsorption of CO_2_ on Activated Carbon with Different Burn-Off

The adsorption isotherm of CO_2_ obtained for LACO, LAC1, and LAC2 at 273 and 300 K are shown in Figure 3. The experimental data demonstrates the same pattern, with an abrupt increase in the isotherms at low pressures and a gradual increase at higher adsorbed pressures. This is a typical isotherm for many micropore adsorbents at close to unity relative pressure; it may rise if mesopores are present due to capillary condensation in the mesopore [27]. As shown in this picture, the adsorption capacities of LACO and both LACs diminish as the temperature increases because of physical adsorption.

The data indicate that the LACs have a higher adsorption capacity than LACO at both adsorption temperatures of 273 K and 300 K, presumably because activated carbon with burn-off has a better porosity ratio or a larger adsorption surface area. Because of its larger BET surface area, as noted in earlier studies, LAC2 adsorbs more CO_2_ than LAC1 when the effect of different percentages of burn-off is considered at 273 K, but at 300 K their adsorption capacities are nearly equal.

#### 2.1.3. Adsorption of CO_2_ and CH_4_ on Activated Carbon Containing Hydroxyl Groups

Figure 4 shows the experimental adsorption isotherms of CO_2_ by longan seed activated carbon with no burn-off (LACO) and with hydroxyl groups (LACM) at 273 and 300 K for the pressure range up to about 5000 mbar. It is obvious that the shape of the isotherms show an initial section of a type I isotherm, according to the IUPAC classification [28,29], at both 273 K and 300 K. Since the amount of CO_2_ adsorbed increases with increasing pressure, the adsorption occurs initially at the smallest pores at low pressure, and then at the next larger pore at a higher pressure via a pore filling mechanism [30]. Many of the pores in LACO and LACM are on the microscopic scale, and the average diameter of a pore is significantly larger than the size of a CO_2_ molecule. Clearly, when comparing temperatures of 273 K and 300 K, CO_2_ adsorption is greater at 273 K, indicating that this adsorption is physical. Then, when comparing LACO and LACM, it can be seen that activated carbon with the addition of hydroxyl groups has a higher CO_2_ adsorption capacity than activated carbon without the addition of any functional groups. This is because the basicity of the hydroxyl group has a positive effect on the adsorption of CO_2_, and the presence of OH on activated carbon can enhance the interaction between CO_2_ and activated carbon.

Figure 5 shows the experimental adsorption isotherms of CH_4_ by longan seed activated carbon with no burn-off (LACO) and with hydroxyl groups (LACM) at 273 and 300 K for the pressure range up to about 5000 mbar. As may be observed, the amount of CH_4_ adsorbed increases with decreasing temperature, proving that the CH_4_ adsorption isotherm represents physical adsorption as well. When the effects of functional groups were compared, it was discovered that activated carbon with no functional groups (LACO) has a higher CH_4_ adsorption capacity than activated carbon containing hydroxyl groups (LACM). In contrast to the illustration in Figure 4, LACM performed better in CO_2_ adsorption. This is due to the adsorbent’s porous structure, which is crucial in adsorption as was previously explained. Even though the average pore diameter of LACO and LACM seems to be larger than that of a CO_2_ molecule, diffusing CH_4_ can be challenging because CH_4_ has a higher kinetic diameter than CO_2_, and LACM may have a narrower average pore width than LACO. In other words, the hydroxyl groups that were added to LACM may cause a pore-blocking phenomenon or interfere with CH_4_ diffusion. Due to this, LACO is able to adsorb more CH_4_ than LACM.

#### 2.1.4. Properties of Activated Carbon with Different Iron Concentrations on Its Surface and Adsorption Isotherms of CO_2_ and CH_4_

Figure 6 shows the results of SEM-EDX analysis of iron-loaded activated carbons for 0 FeAC and 1 FeAC samples. In the position and elemental composition of untreated activated carbon (0 FeAC) and surface-treated activated carbon (1 FeAC), it was discovered that the spectra revealed a carbon atom peak that should be associated with the graphene structure of activated carbon, while iron and oxygen atom peaks should correspond to those of the Fe_2_O_3_ molecule. As in Table 1, which shows the results of an EDX analysis of the elemental composition of activated carbon, the weight% of Fe also increased as the weight% of the solution used for impregnation increased. The amount is quite comparable to what was expected.

Figure 7 shows the experimental excess adsorption isotherms of CO_2_ for a pressure ranging up to 30 bar and those of CH_4_ for a pressure ranging up to 40 bar at a temperature of 273 K on activated carbon with Fe concentrations varying from 0 to 20 wt.%. The adsorption isotherms of CO_2_ and CH_4_ can both be distinguished into two groupings: low Fe content (1–5%) and high Fe content (15–20%). It was discovered that either CO_2_ or CH_4,_ adsorption on activated carbon with a high Fe concentration had the adsorption capacity lower than activated carbon with no additional Fe and activated carbon with a low Fe content. The pore mouth becomes shallow when excessive iron is added. The narrower the pore mouth of activated carbon, the less space there is for the adsorbed gas to be entrapped. For a low Fe content, the adsorption capacity of activated carbon with concentrations of 1 and 5% Fe loading was comparable to that of non-iron-activated carbon (0 FeAC). Activated carbon with a 1% Fe content had a greater adsorption capability. This may be because Fe loading promotes CO_2_ and CH_4_ adsorption capacity by the great interaction between the fluid and ferrous particles. The presence of a low amount of Fe may not narrow the porosity of the activated carbon, however, it can enhance the formation of fluid at the pore mouth and then the diffusion into the inner pore. It is noted that the adsorbed amount of CO_2_ at 273 K on activated carbon derived from longan seeds is greater than that obtained for commercial activated carbon at the same conditions.

### 2.2. Simulation Study for CO_2_ on Perfect and Defective Surfaces

The simulated isotherms versus pressure for CO_2_ at 273 and 300 K in single carbon slit pores with a defective surface of various pore widths (from 6.3 to 30 Å) up to the saturation vapor pressure were obtained by using the GCMC method, as shown in Figure 8 and the simulated isotherms from the perfect surface model are compared in Figure 9.

The simulated carbon dioxide (CO_2_) isotherms in the perfect and defective surface models behave similarly, the isotherms indicate monolayer adsorption with type I isotherms for widths less than 8.5 Å. This is because of the stronger interaction between the fluid and carbons for the narrow pore width. With increasing pore widths, the isotherms become lower, this is due to less potential energy between the fluid and solid. It was observed that at pressure greater than 200,000 Pa the isotherm obtained for 10 Å is greater than that of 8.5 Å, this relates to the packing effect at high pressures. CO_2_ can form a monolayer along the pore wall and then additional layers next to it in the case of pores larger than 8.5 Å. For larger pore widths of 20 to 30 Å, the layering and pore filling mechanisms can be noticed; the isotherm shows a gradual increase in the adsorbed amount due to the formation of a monolayer, followed by a slight increase in the adsorption isotherm at high pressures according to pore filling behavior. In the case of adsorption in larger mesopores, further capillary condensation occurs when the adsorbed phase is dense and then become the liquid phase.

A difference between the adsorbed capacity of carbon dioxide in the perfect and defective surface models is observed in the case of micropores, the adsorbed amount of carbon dioxide on the perfect surface is greater than that on the defective surface. The reason for this comes from the effect of the rotation and orientation of carbon dioxide molecules. The linear model of the three-center LJ model of CO_2_ cannot be packed inside the defect pits; they may lie flat on the perfect surface rather than in the pit because they cannot rotate freely in the narrow pore width, as shown in snapshots in Table 2 and Table 3. However, similar behavior can be observed in the case of the larger pores.

The effects of temperature on the adsorption of CO_2_ for the defective and perfect surfaces obtained for carbon slit pores of 8.5 and 20 Å width are also shown in Figure 10. Some examples of snapshots for CO_2_ inside the pore width of 6.5 Å with the defective surface at various pressures and temperature are also shown in Figure 11 to present the adsorption behavior of CO_2_ molecules on the defective surface. The adsorption of carbon dioxide on carbon is an exothermic process, so, the adsorption isotherm decreases as the temperature rises, as depicted in Figure 10. At higher temperatures, the heat of adsorption is released, and CO_2_ molecules have a high energy of motion. As a result, it is difficult for adsorption to occur on the surface. To put it another way, this is known as physical adsorption.

For a pore width of 6.5 Å, the adsorption on the perfect surface is undoubtedly greater than that on the defective surface, but this is not clearly observed in the case of a 20 Å width. However, when the pressures are greater than 1 MPa, the adsorbed amount of carbon dioxide on the defective surface becomes slightly greater than that on the perfect surface because of the high-pressure adsorption effect; the carbon dioxide molecules can be forced by the pressure to be tightly packed in defect pits on the solid surface.

As discussed above, CO_2_ particles prefer to be adsorbed on the carbon atoms rather than in the defect pits, as shown in Figure 11. The black circles represent carbon atoms, the blue and red circles are those of the carbon and oxygen of carbon dioxide, respectively. We observe the molecules of CO_2_ lie on the carbon surfaces due to the stronger interaction between solid and fluid. The molecular arrangement of CO_2_ on the solid surface is in the linear form and arranged in a horizontal placement rather than a vertical placement. This supports the idea that CO_2_ will be adsorbed in the defect pits at high pressures and this leads to the adsorption capacity of the defective surface becomes greater than the perfect surface.

In Table 2 and Table 3, we also show snapshots of the CO_2_ adsorption mechanism on the perfect and defective surfaces at 273 K, respectively. The three pore widths were chosen as representative of micropores and mesopores. The black spheres represent carbon atoms, the red spheres are carbon atoms of CO_2_, and the blue spheres are oxygen atoms of CO_2_. The slit pore model contains two parallel walls and each wall comprises three graphene layers, for clarity, we show only one graphene layer of each wall. In the case of pores of 6.3 and 6.5 Å widths, a single layer of the molecule can be observed because this pore width can fit with the particle size of CO_2_. The CO_2_ particles lie horizontally parallel to the solid surfaces of both the perfect and defective surfaces. This mechanism leads to a lower adsorption isotherm for the defective surface than the perfect surface. However, at pressures from 100,000 Pa, CO_2_ starts to be adsorbed in the defect pits and becomes fully packed at saturation pressures. This leads to the adsorption isotherm for the defective surface becoming greater than that for the perfect surface at 1 MPa, as observed in Figure 10.

Having seen the adsorption mechanism of CO_2_ in the 6.3 and 6.5 Å width pores with a single layer of particles, now turn to the larger size for micropores of 15 Å and mesopores of 30 Å width for the perfect and defective surfaces in Table 2 and Table 3, respectively. Similar behavior can be observed, first, the monolayer occurs along two pore walls and there is no adsorption at defects. After the formation of the monolayer is complete, then the second layer is forming, as seen at a pressure of 500,000 Pa. In the case of the defective surface, the adsorption at the defect pits can be seen and the pits are then filled with particles at high pressures. For 15 Å width, when the second layer is finished, then the pore filling followed by capillary condensation occurs. For the larger pores, of 30 Å, it is noted that the layering mechanism can form three layers before the pore filling and capillary condensation can be perceived. As one can see in the case of larger pores, CO_2_ can be rotated in any direction not only formed in the horizontal direction during the higher layering mechanism. This may be due to the larger available space meaning that the particles can move independently.

### 2.3. Simulation Study for CH_4_ on Perfect and Defective Surfaces

Figure 12 and Figure 13 show the adsorption isotherms versus pressure for CH_4_ at 273 and 300 K on the perfect surface of various pore widths (from 6.5 to 30 Å) up to 14 MPa, while Figure 14 and Figure 15 show those for CH_4_ at 273 and 300 K on the defective surface obtained by the GCMC simulation. Table 4 and Table 5 show snapshots of CH_4_ particles inside the pores at various pressures for the perfect and defective surfaces, respectively, to reveal the adsorption behavior of the fluid with different pore widths and the effect of defects on the adsorption mechanism. Isotherms can be represented in two distinct ways: linear and semi-log scales. While the semi-log plot is used to compare the adsorption ability of each condition more easily, the isotherms with linear plots serve to demonstrate the type of isotherm.

For simulation isotherms of methane (CH_4_) adsorption on either perfect or defective surfaces of the finite-length carbon slit pore model at 273 and 300 K, the overall observation of the local isotherms of every pore width behave in a similar pattern, where the adsorption branch is concave to the pressure axis, which represents a type I isotherm. Like carbon dioxide adsorption, the layering phenomenon can be observed in the pores having widths less than 8.5 Å, and the layering together with pore filling behavior are observed in the larger pores. It is noted that capillary condensation cannot be observed due to the temperatures being greater than the critical temperature of methane. The adsorbed amount observed from the adsorption isotherm cannot be arranged in ascending order of pore width, from 6.5 to 30 Å, due to the molecular size of the adsorbate gas. The smallest pores in this study is difficult for CH_4_ to move through since CH_4_ is larger than the gap width of 6.5 Å. As a result, the adsorption isotherms for pore having width of 7.0 to 8.0 Å are greater than that of 6.5 Å. The adsorption isotherms for pore width of 8.5 Å and larger pores are less than that of 6.5 Å, due to the less interaction between CH_4_ and carbon atoms.

Comparing the amount of methane adsorbed on the perfect and defective surface models, similarities exist between the adsorption isotherms of the perfect and defective surfaces in that the slopes of the isotherms increase gradually with increasing pressure. As seen by the slopes of the curves, the isotherms of the defective surface are not as sharp or as rapid as those of the perfect surface at low pressures. However, when the pressure was increased, more CH_4_ was pumped into the defect gaps, so the amount of adsorption was comparable. Furthermore, in the case of micropores, the adsorbed amount of methane on a perfect surface is slightly greater than that on a defective surface, but it is quite similar in the case of larger pores.

However, the adsorbed amount of carbon dioxide for the same pore width and temperature is greater than that of methane. The same behavior is also observed in the experiment. This is because carbon dioxide has electrostatic charges from a quadrupole moment that induces greater interactions between carbon dioxide molecules, while methane adsorption is by van der Waals interactions.

To investigate the adsorption mechanism of CH_4_ on the perfect and defective surface slit pores at 273 K, snapshots of particles inside the pores are presented in Table 4 and Table 5, respectively. In these snapshots, the black spheres represent carbon atoms of activated carbon while the blue spheres are methane particles. Micropores and mesopores of 6.5, 15 and 30 Å are chosen such that the adsorption mechanisms of CH_4_ and CO_2_ can be compared. In the case of the 6.5 Å width pore, a single layer of adsorption occurs for both models because the pore space fits one collision diameter of the particles. When the layer is nearly completed, adsorption inside the defect pits occurs, as seen in Table 4, at pressures from 1 MPa. The initial adsorption takes place at the inner pore and then it goes along the 4pore wall to the pore mouth for the perfect surface model, because of the stronger interaction between the fluid and solid at the pore’s center. However, for the defective surface model, the defects are distributed randomly on the surface including the inner pores, then the initial adsorption can be observed at the carbons of the inner graphene layer. The first layer can be formed along the pore wall and then the defects are filled in. This mechanism is also observed for CO_2_ adsorption.

Now, turning to the adsorption mechanism for larger pores of 15 and 30 Å width, the initial adsorption takes place at two pore walls and the first layer can be formed along the pore wall due to the stronger interaction. Then, the pore filling behavior can be observed but the condensation cannot be seen because the adsorption temperature of 273 K is greater than the critical temperature of methane, as mentioned above. Therefore, methane behaves as a supercritical fluid, for which no condensation happens, and this leads to the type I adsorption isotherm for a supercritical fluid. In the case of the defective surface model, CH_4_ particles start to be adsorbed in the pits at 1 MPa and fully fill all the defects at 5 MPa. Therefore, the adsorption behaviors of CH_4_ and CO_2_ show initial adsorption along the pore wall, when the first layer is complete and the second layer is forming, then, adsorption in the defects occurs. The adsorption in the slit pores of the perfect surface is greater than that for the defective surface at pressures lower than 1 Mpa; after that, the opposite behavior can be found. This may lead to a greater adsorption capacity at saturation conditions or at higher pressures.

### 2.4. Effects of Iron Loading on Adsorption Isotherms

Figure 16 depicts the adsorption isotherms versus pressure for CH_4_ at 273 K for finite-length slit pores with various Fe amounts (0–20%) on the surface, whereas Figure 17 depicts the isotherms versus pressure for CH_4_ at 273 K for a carbon slit pore with 1% Fe on its surface and pore widths ranging from 7 to 20 Å. A semi-log plot is used to represent the mechanism and the adsorption capacity, whereas a linear plot is used to evaluate the types of isotherms.

Adsorption occurs extremely quickly at low pressures, as seen in Figure 16. The surfaces with low Fe concentrations (1–5%) have a high adsorption capacity for methane, which is in line with experimental observations for Fe at a concentration of 1% on an activated carbon surface. When compared to the activated carbon without Fe added to the surface, the pore with 1% Fe content has a greater capability for methane adsorption. However, the pore with 15% Fe on the surface has poor methane adsorption. This is because an excessive amount of iron restricts the pores, which in turn prevents methane particles from being adsorbed on the surface.

Upon comparing methane adsorption with different pore widths for graphene with 1% Fe on the surface, as seen in Figure 17, it was discovered that a pore with a width of 7 Å had the greatest methane adsorption capacity, followed by those with widths of 10, 15, and 20 Å, in that order. It is obvious that 7 Å pores adsorb methane more quickly than other pores at low pressure. This is because narrow pores have a greater force of attraction than wider ones, i.e., large pores have less attraction between the surface and the methane. A pore width of 7 Å is particularly advantageous for methane adsorption because it allows the gas to be packed densely with minimal gaps.

### 2.5. Effect of Hydroxyl Groups (OH) on CO_2_ Adsorption

Having seen the effect of iron on the adsorption isotherms, now we examine how functional groups influence the carbon dioxide capture in porous carbon at 273 K. The adsorption isotherms of CO_2_ with 5% OH groups located randomly for 6.5, 7.0 and 15 Å pore widths using the GCMC simulation method are shown in Figure 18. For comparison, the adsorption isotherms obtained for the same pore widths in the absence of the functional groups are also plotted in the same figure. The isotherms at low pressures (*p* < 1 kPa) show a less significant effect for functional groups on the adsorbed amount, however, this effect becomes significant at higher pressures for smaller pores of 6.5 and 7 Å. The isotherm for 5% OH groups is greater than that for a homogeneous pore at the same pore size, this is due to the greater electrostatic force between CO_2_ and the functional groups. However, the isotherms of carbons with functional groups become less than the homogeneous surface carbon at pressures greater than 600 kPa. This may be because the inner pore space is filled with functional groups, reducing the pore volume for CO_2_ capture. This behavior is similar to that observed in the experimental data compared between LACO and LACM in Figure 4. An insignificant effect for the larger pore of 15 Å is seen and this could imply that the smaller pore width together with the functional groups can enhance carbon dioxide capture.

## 3. Methodology

### 3.1. Computer Simulation

#### 3.1.1. Fluid Model

Carbon dioxide is modeled as a 3-center Lennard-Jones (LJ) molecule with LJ interaction sites on the atoms and point charges to account for the quadrupole moment, whereas methane is modeled as a simple spherical LJ molecule. The porous carbon model is assumed to be a parallel pair of finite-length graphene layers composed of carbon atoms aligned in a hexagonal arrangement. The molecular parameters for CO_2_ and CH_4_ are shown in Table 6. The Lennard-Jones 12-6 equation is used to compute the interaction energy between two LJ sites, while Coulomb’s law of electrostatic interaction is used to calculate the interaction energy between two charges [31].

#### 3.1.2. Perfect Surface Model

The carbon-based adsorbent’s structure with pores with a slit-shaped geometry was chosen as a solid model in this research. A simple finite-length slit pore is modeled as a parallel pair of finite-length graphite layers perpendicular to the *z*-axis. Each of the two walls is composed of three graphene layers that are placed on top of one another with a 3.354 Å interlayer gap. The width, H, of this slit pore model is defined as the distance between a plane running through all carbon atom centers of one wall’s outermost layer and the corresponding plane of the opposite wall. Each layer’s carbon atoms are arranged in the form of a condensation of aromatic rings composed of six carbon atoms, with a neighboring carbon–carbon spacing of 1.42 Å [21].

#### 3.1.3. Defective Surface Model

In the reality of porous carbon as described above, activated carbon also has defect pits on its basal graphene layers [35]. So, this study also observes the effect of defective surfaces with the finite length of the carbon slit pore model on the adsorption behavior of carbon dioxide and methane. To construct a defect in a surface model, an atom of carbon is chosen at random and then removed from the surface, as well as all of its neighbors that are closer to it than the effective defect radius, R_c_. This process will keep going until the percentage of carbon atoms that are removed reaches a certain value. The percentage of defects and the size of the defects, which is determined by the effective radius, are the two most crucial parameters for modeling a defective surface or a non-graphitized surface in a surface modeling system [35]. The selected values of these two parameters are 29% for the defect percentage and 4.92% for the defect size or effective radius. These selected values give the best agreement between the simulation results and the experimental data of carbon adsorption. If other adsorbents are to be studied, these two parameters should be varied in the simulation study with suitable values selected that agree well with the experimental data of the adsorbent. An example of defective surface creation is shown in Figure 19.

#### 3.1.4. Functional Group Model

The functional groups on activated carbon surfaces have been identified as hydroxyl, carbonyl, carboxylic, phenolic, lactonic, and pyrone, however, to simplify the model, hydroxyl groups (OH) are considered to represent the surface functional groups, as in Muller et al. [18]. The center of oxygen is the LJ dispersive site of the OH group, which has a negative charge of −0.64 e. The distance between the dispersive site and the pore wall is 1.364 Å when the dispersive site is perpendicular to the wall. The center of hydrogen has a positive charge of +0.45 e and is approximately 0.96 Å away from the functional group’s center. The OH groups’ molecular parameters used in this study were ɛ_ss_/k_b_ of 78.23 K, σ_ss_ of 3.07 Å, and the angle of O-H of 109° [34,36]. The surface chemistry properties of activated carbon used in this study were measured for the total amount of acidic and basic groups by Boehm titration.

#### 3.1.5. Iron Oxide Model

Iron oxide is used as the impurity on the activated carbon surface, and its LJ dispersive site is located 1.45 Å [37] away from the surface, with a collision diameter of 2.3193 Å and an energy well depth of 6026.7 K [38]. The iron loading on the graphene layer is shown in Figure 20.

#### 3.1.6. Grand Canonical Monte Carlo (GCMC) Simulation

In the Monte Carlo (MC) simulation, the metropolis algorithm is used [31], and the GCMC ensemble is used to determine the CH_4_ and CO_2_ adsorption isotherms. This ensemble’s simulation box is a finite-length carbon slit pore with a linear dimension of 60 Å in both the x and y directions. The top and bottom of the simulation box are assumed to be two slit pore walls, with each wall consisting of three graphene layers. To achieve the adsorption equilibrium, we specify the volume of the box (i.e., pore volume), the chemical potential, and the temperature of the system. One GCMC cycle consists of a thousand displacement moves and equally probable insertion and deletion attempts. An adsorption branch of the isotherm generally takes 30,000 cycles to attain equilibrium, with an additional 30,000 cycles required to establish ensemble averages. We use an empty box where the amount of fluid specified in the simulation box is equal to zero, as the initial configuration for each point on the adsorption branch, and the simulation is run until the number of particles in the box does not change (in the statistical sense). The equation of state established by Johnson et al. is used to compute the pressure of the bulk gas corresponding to a given chemical potential [39].

### 3.2. Experimental Work

The experimental adsorption isotherm was determined using longan seed activated carbon with varying degrees of burn-off (%), 19% for LAC1, 26% for LAC2, and no burn-off for LACO. Moreover, LACM is an acronym for longan seed activated carbon contained hydroxyl groups. Using an intelligent gravimetric analyzer (IGA), model IGA-002 supplied by Hiden Analytical Ltd., Warrington, UK, the experimental data for CO_2_ and CH_4_ adsorption in LACO, LAC1, and LAC2 at 273 and 300 K were acquired. An adsorption experiment begins with a 10 h outgassing of a 0.12 g carbon sample at 200 °C, followed by a cooling period to the adsorption temperature.

When referring to activated carbon that has different amounts of iron on its surface, the notation xFeAC is used, where x indicates the percentage by mass. The typical impregnation approach used to create Fe_2_O_3_ supported on activated carbon originated from Hakim et al. [15]. The findings of CO_2_ and CH_4_ adsorption studies on xFeAC at 273 K were obtained using Micromeritics’ high-pressure volumetric analyzer (HPVA II). To eliminate any impurities still present in the pores of the adsorbent, the 1 g or so of xFeAC sample was placed into the sample tube and degassed for 12 h at 573 K (300 °C). Following that, the sample tube was moved to the measurement port to gather adsorption data at a constant temperature. The elemental identification and quantitative compositions of the activated carbon samples were analyzed using a scanning electron microscope with energy dispersive X-ray (SEM-EDX) instrument (JSM-7800F, JEOL).

## 4. Conclusions

In the experimental part, the adsorption capacities of carbon dioxide on burned-off activated carbons (LACs) and activated carbons with hydroxyl groups added (LACM) were greater than on original activated carbons (LACO) derived from longan seeds. For adsorbing methane, LACO performed better than LACM, and the excessive addition of iron to the activated carbon surface was deleterious to the adsorption capacity. The adsorption of CO_2_ at 273 K in LACO was about 4 mmol/g at 1000 mbar and it increased to 4.2 and 4.8 mmol/g for 19 (LAC1) and 26% (LAC2) burn-off activated carbons, respectively. This is due to an increase in the porous properties of the BET area, micropores, and surface chemistry. In the investigation of the impact of hydroxyl groups on the adsorption of CO_2_ and CH_4_ at 273 K, it is found that the maximum adsorption capacity of CO_2_ at 1000 mbar is about 4 and 4.6 mmol/g in the absence (LACO) and presence of hydroxyl groups (LACM) contained on the carbon surfaces. This is due to the greater interaction between CO_2_ and OH than that between fluid and the carbon surface. Compared to the other activated carbon, derived from bamboo with a higher degree of burn-off, it was found that the activated carbon with a greater degree of burn-off provided a greater micropore volume and BET area. Therefore, we have developed the OTA method [40,41] to produce a high BET surface area of 1773 cm^2^/g, micropore volume of 0.600 cm^3^/g, and mesopore volume of 0.474 cm^3^/g.

In the research of the adsorption based on the molecular simulation, when comparing the influence of pore width, the adsorbed amount is greater for micropores than for larger pores. The adsorption capacity of CO_2_ on a perfect surface is greater than that on a defective surface at pressures lower than 1 MPa because of the molecular arrangement of CO_2_ on carbons and because there is no adsorption in the defects. Whereas at pressures greater than 1 MPa, pore filling behavior occurs and leads to the adsorption capacity of CO_2_ on defective surface increasing faster than on a perfect surface. Due to a stronger mutual attraction, carbon dioxide can be adsorbed in greater amounts than methane. Additionally, adding iron to the surface exhibited the same results as the experiment, demonstrating the negative effects of excessive iron content. Therefore, the simple model of finite-length slit pores can be used to describe the adsorption mechanism of fluid in activated carbon with different degrees of burn-off and surface chemistry quite well.

## Figures and Tables

**Figure 1 molecules-28-05433-f001:**
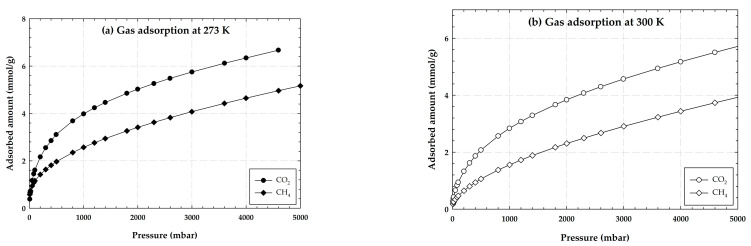
Experimental adsorption isotherms of CO_2_ and CH_4_ on LACO at 273 and 300 K.

**Figure 2 molecules-28-05433-f002:**
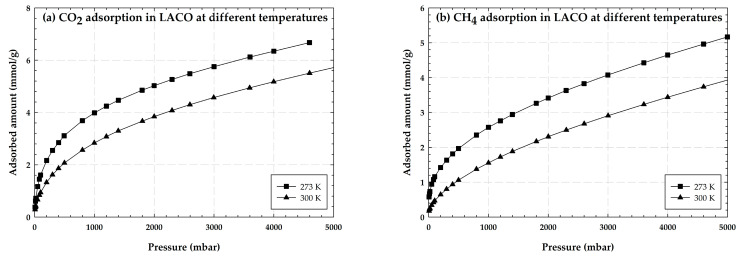
Experimental adsorption isotherms of CO_2_ and CH_4_ on LACO at different temperatures.

**Figure 3 molecules-28-05433-f003:**
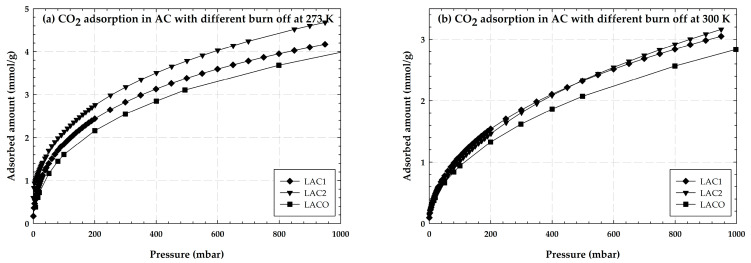
Experimental adsorption isotherm of CO_2_ on activated carbon with different burn-off at 273 and 300 K.

**Figure 4 molecules-28-05433-f004:**
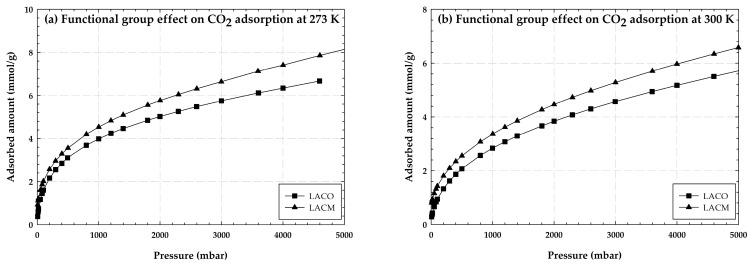
Experimental adsorption isotherms of CO_2_ on LACO and LACM at 273 and 300 K.

**Figure 5 molecules-28-05433-f005:**
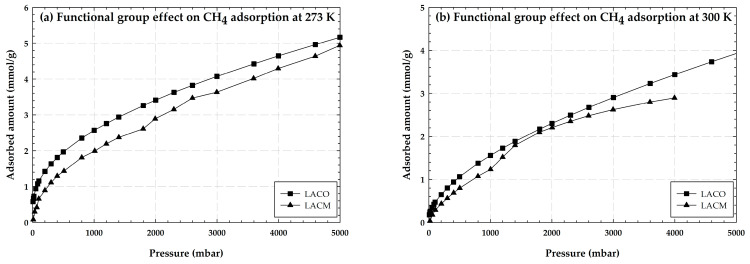
Experimental adsorption isotherms of CH_4_ on LACO and LACM at 273 and 300 K.

**Figure 6 molecules-28-05433-f006:**
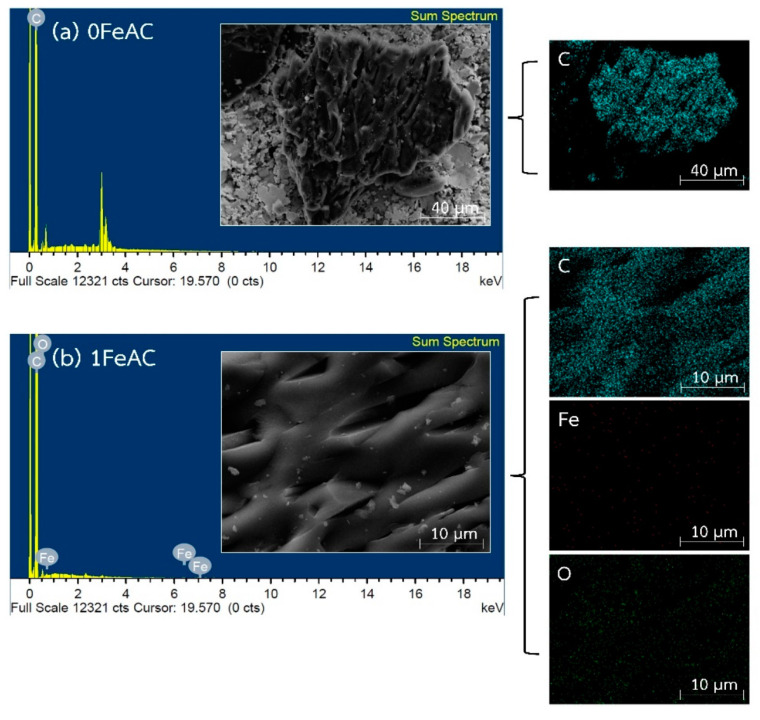
Results of SEM-EDX analysis of metal loading activated carbons (0 FeAC and 1 FeAC samples).

**Figure 7 molecules-28-05433-f007:**
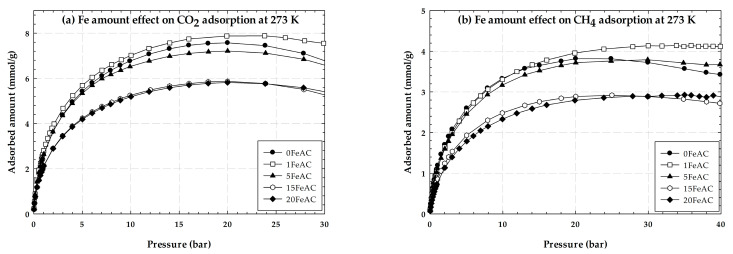
Experimental excess adsorption isotherms of CO_2_ and CH_4_ on activated carbon with different Fe at 273 K.

**Figure 8 molecules-28-05433-f008:**
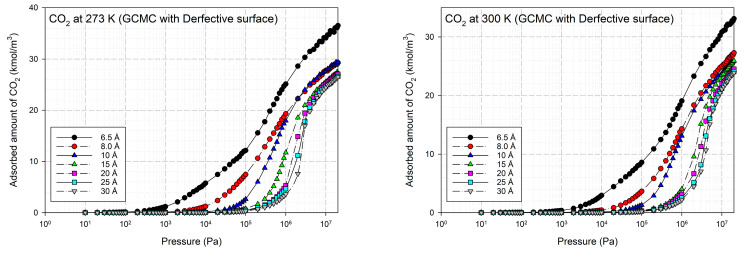
Simulated adsorption isotherms of CO_2_ for defective carbon surface at 273 and 300 K at various pore widths from 6.5–30 Å.

**Figure 9 molecules-28-05433-f009:**
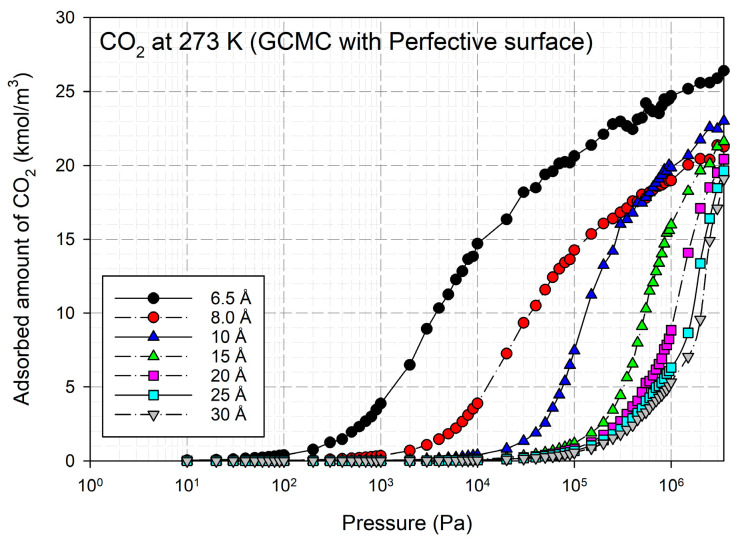
Simulated adsorption isotherms of CO_2_ for perfect carbon surface at 273 K at various pore widths from 6.5–30 Å.

**Figure 10 molecules-28-05433-f010:**
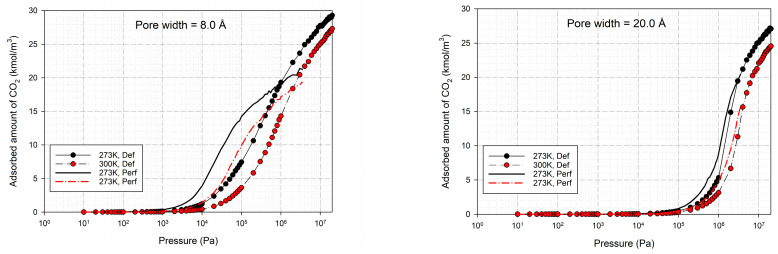
Simulated adsorption isotherms of CO_2_ at different temperatures for pore widths of 8.5 and 20 Å: comparison between the defective (Def) and perfect (Perf) carbon surfaces.

**Figure 11 molecules-28-05433-f011:**
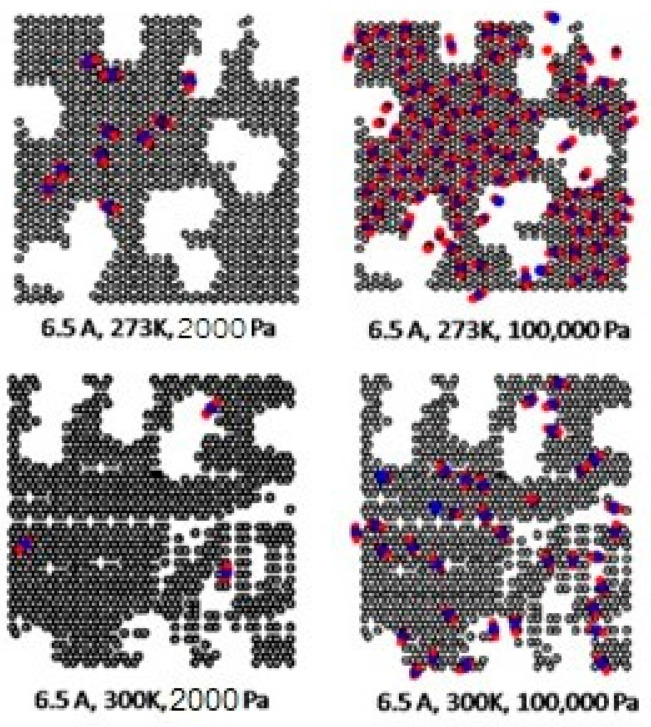
Snapshots of CO_2_ at 273 and 300 K in a pore width of 6.5 Å at various pressures and temperatures.

**Figure 12 molecules-28-05433-f012:**
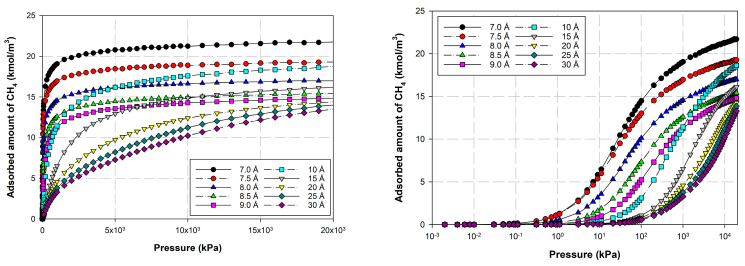
Simulation isotherms of methane in carbon slit pore model with a perfect surface at 273 K with different pore widths.

**Figure 13 molecules-28-05433-f013:**
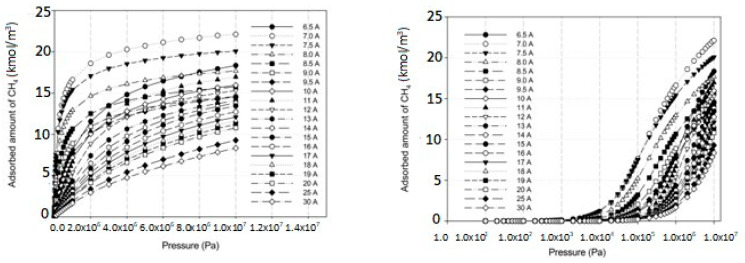
Simulation isotherms of methane in carbon slit pore model with a perfect surface at 300 K with different pore widths on linear and semi-log scales.

**Figure 14 molecules-28-05433-f014:**
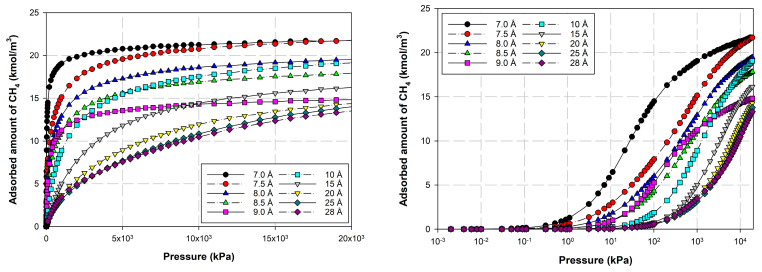
Simulation isotherms of methane in carbon slit pore model with a defective surface at 273 K with different pore widths on linear and semi-log scales.

**Figure 15 molecules-28-05433-f015:**
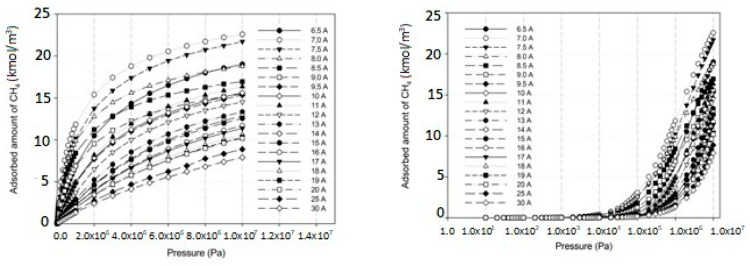
Simulation isotherms of methane in carbon slit pore model with a defective surface at 300 K with different pore widths on linear and semi-log scales.

**Figure 16 molecules-28-05433-f016:**
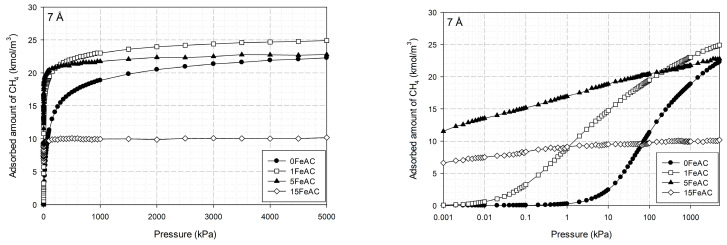
Adsorption isotherms of methane in the carbon slit pore model at 273 K with varying concentrations of Fe on its surface.

**Figure 17 molecules-28-05433-f017:**
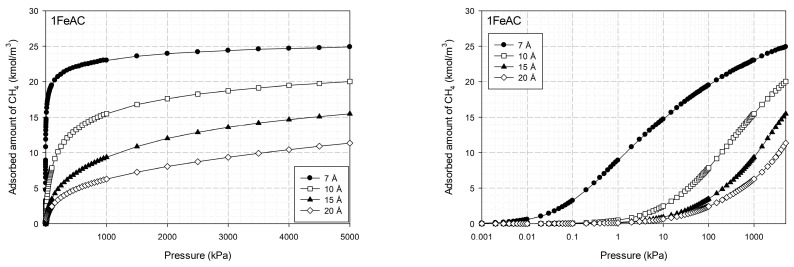
Adsorption isotherms of methane in the carbon slit pore model with 1% Fe on its surface at 273 K for various pore widths.

**Figure 18 molecules-28-05433-f018:**
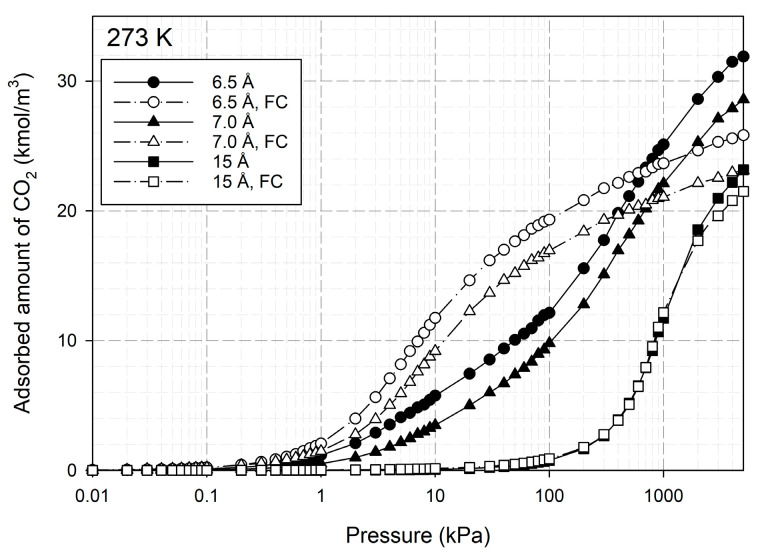
Adsorption isotherms of carbon dioxide at 273 K for different pore widths in the presence and absence of hydroxyl functional groups (FC).

**Figure 19 molecules-28-05433-f019:**
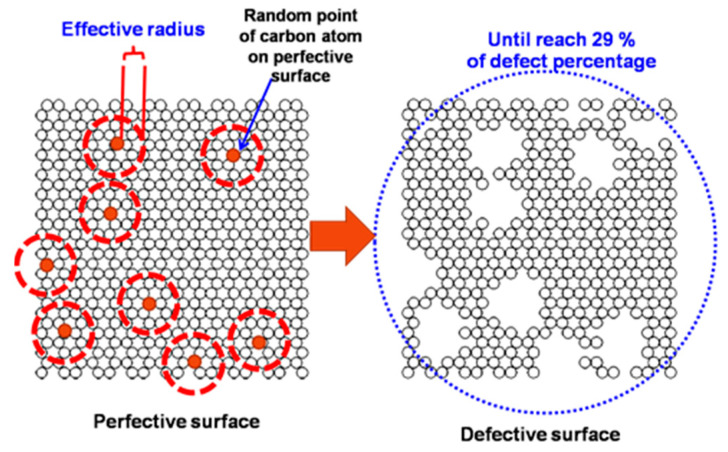
The creation of a defective surface on an outermost graphene layer.

**Figure 20 molecules-28-05433-f020:**
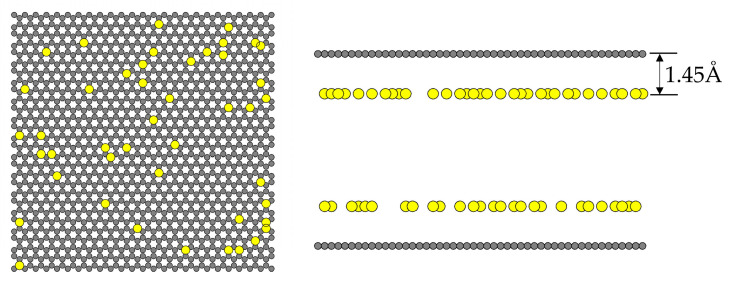
The iron loading on the graphene layer.

**Table 1 molecules-28-05433-t001:** Elemental compositions of iron-loaded activated carbons obtained from SEM-EDX analysis.

Sample	Composition (wt.%)		
	Fe	O	C
0 FeAC	-	-	100
1 FeAC	0.65	2.63	96.72
5 FeAC	3.12	8.85	88.03
15 FeAC	14.59	13.89	71.51
20 FeAC	20.48	16.83	62.69

**Table 2 molecules-28-05433-t002:** Snapshots of CO_2_ adsorbed in finite-length carbon slit pore with perfect surface model using GCMC simulation at 273 K.

Pore Width 6.3 Å	Pore Width 15 Å	Pore Width 30 Å
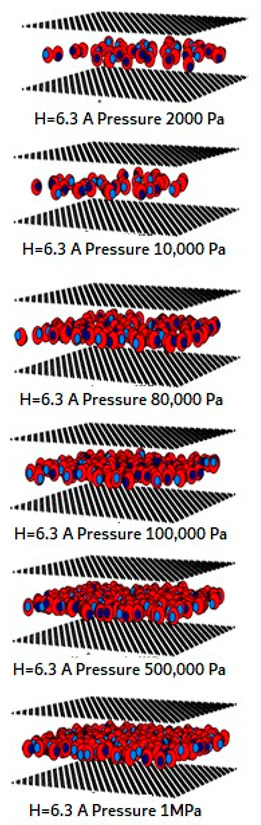	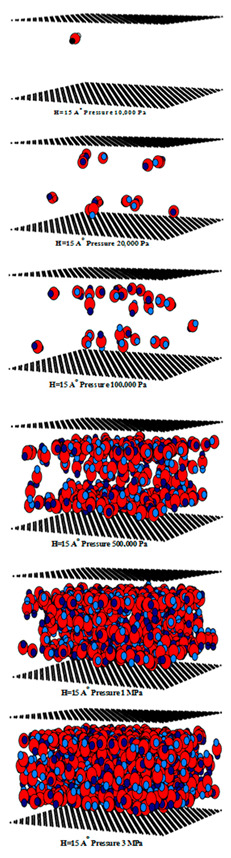	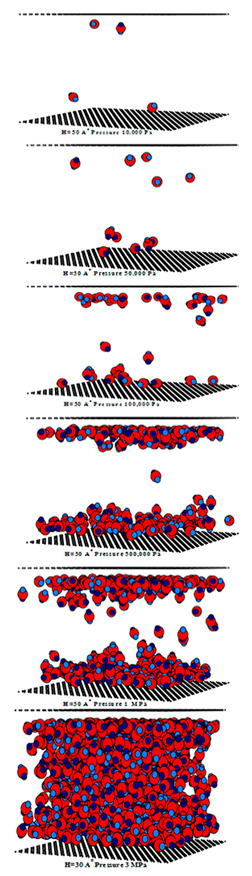

**Table 3 molecules-28-05433-t003:** Snapshots of CO_2_ adsorbed in finite-length carbon slit pore with defective surface using GCMC simulation at 273 K.

Pore Width 6.5 Å	Pore Width 15 Å	Pore Width 30 Å
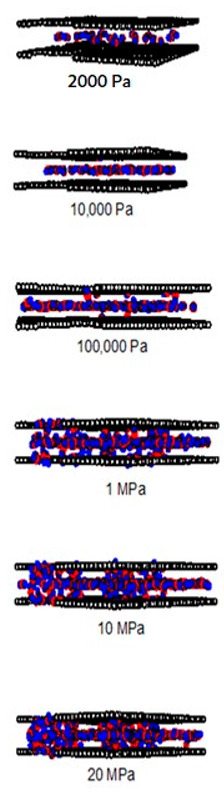	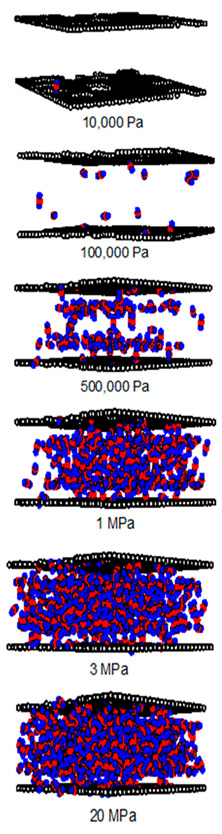	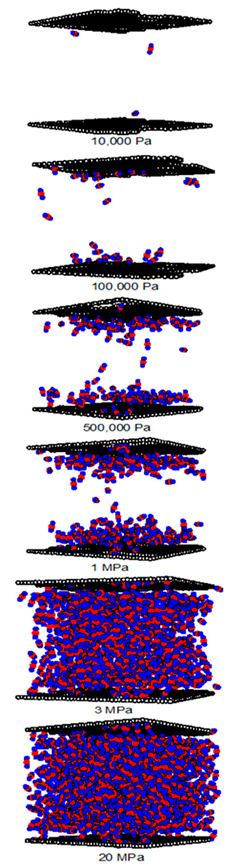

**Table 4 molecules-28-05433-t004:** Snapshots of CH_4_ in the finite-length carbon slit pore model obtained from the GCMC simulation at 273 K.

Pore Width 6.5 Å	Pore Width 15 Å	Pore Width 30 Å
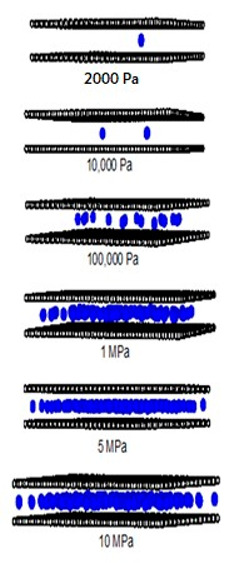	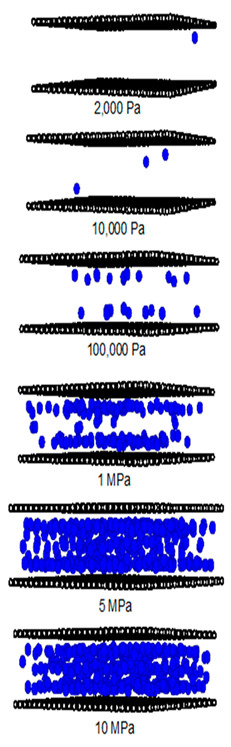	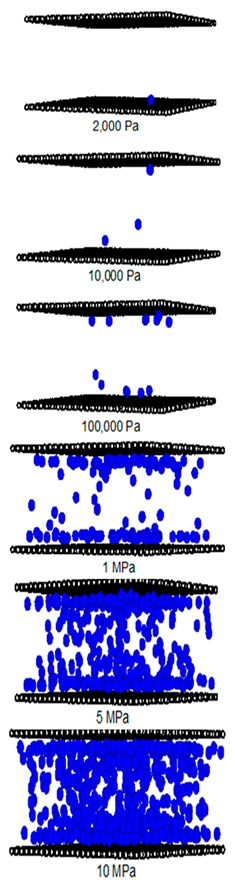

**Table 5 molecules-28-05433-t005:** Snapshots of CH_4_ in the finite-length carbon slit pore with defects model obtained from the GCMC simulation at 273 K.

Pore Width 6.5 Å	Pore Width 15 Å	Pore Width 30 Å
	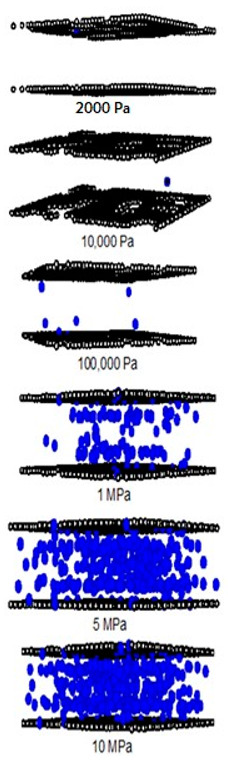	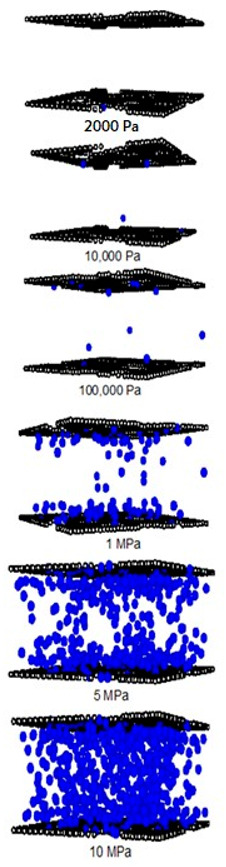

**Table 6 molecules-28-05433-t006:** Molecular parameters and partial charges for CO_2_ and CH_4_ used in this study.

Type	Interaction Site	Collision Diameter, σ (Å)	Energy Well Depth, ε/k_b_ (K)	Atomic Charge, q (e)	References
CO_2_	C	2.757	28.129	+0.6512	[32]
(Multi-site model)	O	3.033	80.507	−0.3256	
CH_4_	CH_4_	3.73	148.0	0	[33,34]
(Spherical model)					

## Data Availability

The data presented in this study are available upon request.

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
