# Peer review of "The Enhancement of CO2 and CH4 Capture on Activated Carbon with Different Degrees of Burn-Off and Surface Chemistry"

_molecules, 2023, doi:10.3390/molecules28145433_

Round 1

Reviewer 1 Report

The article ‘Effects of Surface Heterogeneity on CO2 and CH4 Capture on Activated Carbon: Experiment and Simulation’’ is interested article, however, it require few changes. Moreover the technical, novel side of the paper is very week. Following are the comments for the authors to improve the article:

Ø  Ø  The article title is good statement, however, authors are suggested to re-write the title with the research impact and novelty perspective.

Ø  Ø  A lot of research is going on Surface Heterogeneity on CO2 and CH4 Capture for different adsorbents. How authors claim the new aspect for the need of this publication?

Ø  Ø  Specific problem statement should be mentioned at the start of the abstract, it is missing. Why the study is important?

Ø  Ø  What is the base of selection of this study? Author should define some specific criteria.

Ø  Ø  Cumulative references should be avoided i.e   there should be only 1-2 references. Instead of merging references, authors should add more up to date and state of the art literature, if possible.

Ø  Ø References [7-8] repeat. Please recheck.

Ø  Ø  Line 82, Table 1, it is mentioned ‘’Results’’ please check. Also define parameters description in Table 1.

Ø  Ø  There should be some statistical figured values in the abstract which can quantify the research / optimization and it can make readership of the journal easy.

Ø  Ø  The abstract should also include the solution of the problem based on the problem statement with some particular application/s.

Ø  Ø  There are few old references, authors are encouraged to add latest literature.

Minor spell check etc. 

Author Response

Dear the reviewer

Thank you for your comment, we would like to submit our response for your consideration as the attachment. 

Best regards

Atichat Wongkoblap

Reviewer 2 Report

Comments from Reviewer

Title: Effects of Surface Heterogeneity on CO2 and CH4 Capture on Activated Carbon: Experiment and Simulation

The current form's presentation of methods and scientific results is unsatisfactory for publication in the Molecules journal. The minor and significant drawbacks to be addressed can be specified as follows:
1.    I suggest writing an abstract with descriptive and concise information about this research. A standard abstract definition is: "An abstract is a concise summary of an experiment or research project. It should be brief -- typically under 200 words. The purpose of the abstract is to summarize the research paper by stating the purpose of the research, the experimental method, the findings, and the conclusions."
2.    Please do not use abbreviations in the abstract. The abbreviation GCMC was used only once in the abstract, so there is no point in introducing it.
3.    Lines 2, 10 and others, CO2 and CH4. Subscript!!!
4.    Lines 42 and 43, [7-8]. [7-8] --> [7, 8]. It is necessary to check the entire manuscript. See, for example, line 56.
5.    Lines 53-68. The authors should clearly define the study's objectives and indicate the elements of scientific novelty.
6.    Line 82. “2. Results.”????????????????
7.    Tab. 1. Superscript???? Lines 77 and 78 – the lack of subscripts.
8.    2.1.3. Defective surface model. Please provide me with literature reports confirming the existence of such defects in the walls of carbon materials.
9.    Line 119. Why do the introduced iron atom authors call "functional group"?
10.    Figs. 2-6. Why do black areas appear in these figures?
11.    3.1.3. Adsorption of CO2 and CH4 in activated carbon contained hydroxyl group. How did the authors estimate the number and type of surface groups? Please publish these results!
12.    3.1.4. Adsorption isotherms of CO2 and CH4 in Activated carbon with different Fe contents. The same questions as for No. 11 (see above).
13.    Fig. 10 is illegible.
14.    Page 12. Empty boxes?????
15.    Tab.3. Why is the wall made of one sheet? Membranes?
16.    Figs. 16 and 17. These figures were not properly prepared either.
17.    4. Cocnlusions. The authors did not conduct any discussion based on earlier published papers. Hundreds of papers on CO2 and CH4 adsorption have been published. Both theoretical and experimental ones. There is no answer to the questions of whether their carbon materials adsorb both adsorbents better compared to other materials. What is the adsorption capacity of the studied pores with models with more realistic and complex structures, for example, kerogen.
18.    Literature should also be standardized: the size of letters in the titles of journals, initials of names, the size of letters in the titles of articles. See, for example, (i) Line 607: “J. Chem. Eng.” and Line 611: “The Journal of Physical Chemistry” (ii) Ref. [27]? no bibliographic data

Sincerely,
    The reviewer.

Author Response

Dear the Reviewer

Thank you for your suggestion, I would like to submit my response for your consideration.

Best regards,

Atichat Wongkoblap

Round 2

Reviewer 2 Report

Congratulations on a great job. The author has made a substantial improvement for this article. The manuscript can be accepted for publishment in the present form.